# The Versatile Role of Matrix Metalloproteinase for the Diverse Results of Fibrosis Treatment

**DOI:** 10.3390/molecules24224188

**Published:** 2019-11-19

**Authors:** Hong-Meng Chuang, Yu-Shuan Chen, Horng-Jyh Harn

**Affiliations:** 1Buddhist Tzu Chi Bioinnovation Center, Tzu Chi Foundation, Hualien 970, Taiwan; kavin273@gmail.com (H.-M.C.); yushuanchenxie@gmail.com (Y.-S.C.); 2Department of Medical Research, Hualien Tzu Chi Hospital, Hualien 970, Taiwan; 3Department of Pathology, Hualien Tzu Chi Hospital & Tzu Chi University, Hualien 970, Taiwan

**Keywords:** matrix metalloproteinase, extracellular matrix, fibrosis

## Abstract

Fibrosis is a type of chronic organ failure, resulting in the excessive secretion of extracellular matrix (ECM). ECM protects wound tissue from infection and additional injury, and is gradually degraded during wound healing. For some unknown reasons, myofibroblasts (the cells that secrete ECM) do not undergo apoptosis; this is associated with the continuous secretion of ECM and reduced ECM degradation even during de novo tissue formation. Thus, matrix metalloproteinases (MMPs) are considered to be a potential target of fibrosis treatment because they are the main groups of ECM-degrading enzymes. However, MMPs participate not only in ECM degradation but also in the development of various biological processes that show the potential to treat diseases such as stroke, cardiovascular diseases, and arthritis. Therefore, treatment involving the targeting of MMPs might impede typical functions. Here, we evaluated the links between these MMP functions and possible detrimental effects of fibrosis treatment, and also considered possible approaches for further applications.

## 1. Introduction

Matrix metalloproteinases (MMPs) are endopeptidases with a Zn^2+^ ion catalytic domain [1]; they interact with multiple components of the extracellular matrix (ECM) and bioactive molecules such as receptors and cytosolic phosphatase [2,3]. Novel substrates of MMPs are still being identified, such as cytokines and growth factors [4]. The classification of MMPs is based on the substrate that they degrade, while the naming is not specific to the catalytic activity [5,6]. For example, MMP-1, also known as collagenase 1, can digest Col I, II, III, VII, VIII, X, and gelatin [7]. Subsequently, membrane-type MMPs (MT-MMPs) were discovered, which have a transmembrane domain from the extracellular to the cytosolic part of the cell [8]. There are other membrane-anchored metalloproteinases with a disintegrin domain, which belong to two new families, referred to as the ADAMs (A Disintegrin And Metalloproteinases) and ADAMTs (A Disintegrin And Metalloproteinases with Thrombospondin Motifs) [9]. The inhibitory pro-domain and the zinc-binding catalytic domain are the central features of MMPs, and domains corresponding to these are also present in ADAMs and ADAMTs, which have a cysteine-rich domain, epidermal growth factor (EGF)-like domain, and type-1 thrombospondin (TSP-1) domain [10]. These domains indicate that the key function of ADAMs is in the ectodomain shedding of membrane proteins, although some ADAMs can also degrade ECM substrates. The most intensively studied ADAM is ADAM17, which facilitates the release of the soluble form of tumor necrosis factor-α (TNF-α) from its membrane precursor. Unlike studies of the most critical biological functions of ADAMs on MMPs, there have been fewer studies on the use of ADAMs for ECM degradation. As such, this work focuses on the experimental evidence of using MMPs as targets in studies of organ fibrosis.

Given that their catalytic activity is specific to conserved collagen-like peptides, MMPs have often been linked to fibrosis and cancer metastasis [11]. The roles of MMPs in fibrogenesis are linked to an imbalance between ECM secretion and MMP degradation [12,13]; in tumor metastasis, MMPs degrade cell–cell junctions, which promotes invasiveness into adjacent tissues [14,15]. Therefore, the regulation of ECM-degrading enzymes may be a rational therapeutic target in both fibrosis and tumor metastasis [16,17]. Although most studies have shown that disruption of the activity or expression of MMPs reduced fibrosis, Giannandrea and Parks have listed the diverse treatment results for fibrosis in different types of MMPs [18]. Moreover, the contradictory roles of MMPs have been reported not only in the tumor microenvironment, but also in relation to the acquisition of properties for cancer growth and invasion [19]. Thus, cellular physiology or tissue homeostasis might change when targeting MMPs to treat organ fibrosis. For instance, MT1-MMP cleavage activates MMP2, thus maintaining its activity even in the presence of tissue inhibitors of metalloproteinases (TIMPs) and causes ECM remodeling [20]. Moreover, activated MMPs enhance EMT in epithelial cells, resulting in transformation of the cell type [21,22].

Interestingly, the expression of MMPs was elevated in the early stage of fibrosis, even before the accumulation of scar tissue, and they were reduced after the recovery stage [23]. It is believed that MMPs play an important role that could be inhibited to treat fibrosis. Notably, the results suggested a diverse therapeutic effect of MMP targeting. Here, we discuss the general and correlated functions of MMPs that might alter the treatment of fibrosis. Moreover, MMPs are also related to cancer, cardiovascular, and nervous system diseases. Based on the possible significance of MMPs for treating fibrosis, but also the uncertainty about their therapeutic potential, the possible mechanisms of action of MMPs are discussed in this review, and hypotheses are proposed about the roles of MMPs in fibrogenesis and its therapy.

## 2. General Functions and Regulation of MMPs

The endopeptidase activity of MMPs is derived from their catalytic domain, which is inhibited by the pro-domain (consisting of the conserved amino acid sequence PRCGXPD) [24]. Thus, the general MMP is secreted in a latent form and located depending on its domain-property; as such, the transmembrane domain-containing MT-MMPs act as membrane proteins. Some MMPs are not secreted and instead perform different functions in the cytoplasm. For example, as a regulator of cellular communication network factor 2 or connective tissue growth factor (CCN2/CTGF), MMP-3 plays a role in the nucleus, which is due to its attenuated signal peptide at the N-terminus [25]. Moreover, MMP-2 has been shown to be present in the cytosol of cardiomyocytes, due to its cleavage by troponin I, during ischemia–reperfusion injury [26,27]. However, the Human Genome Project revealed that the real gene numbers are far fewer than those that are predicted [28], which suggests that some protein-coding genes, including MMPs, may have more undiscovered functions.

### 2.1. The ECM Digestion Processes of MMPs

All MMPs catalyze the breaking of peptide bonds via their Zn^2+^-containing domain. Some of them, such as MMP-2 and MMP-9, contain a fibronectin-like region for more inseparable binding to their substrate [29,30]. The digestion process of MMP-3 was described by Pelmenschikov et al. [31,32]. Except for the catalytic domain, members of the MMP family have some motifs such as the N-terminal signal peptide that confer the optional secretory property of MMPs [33]. The hinge region contains a proline-rich region and cooperates with the hemopexin-like C-terminal domain that consists of a four-bladed β-propeller structure. This interacts with TIMP-1 and inhibits the activity of pro-MMP-9 and cell migration [34,35]. Furthermore, the hinge and hemopexin-like region form a proline zipper-like structure to unwind the triple helix of collagen [36,37] because the substrate-binding site is too narrow (about 5 Å) from the triple-helix-typed structure of collagen [38]. It is now clear that MMPs are very effective in catalyzing specific substrates like collagen and gelatin.

### 2.2. The Regulation of MMPs Corresponding to Physiological Processes

MMPs play major roles in cell development and migration because of their ECM-degrading activity, which is controlled by triggers such as growth factors or cytokines acting on cis-elements, including activator protein 1 (AP-1) and polyoma enhancer activator 3 (PEA3) at the promoter upstream [39]. For the wound-healing response, keratinocyte migration, angiogenesis, and contraction are correlated with MMP-13; thus, MMP-13 knockout mice exhibit impaired wound-healing response [40]. In skin diseases, epidermal growth factor (EGF)-induced MMP-1 expression in skin fibroblasts has been shown to be related to the deregulation of matrix metabolism [41]. Furthermore, studies have revealed that matrix stiffness regulates MMP-9 and TIMP-1 to perpetuate fibrosis in hepatic stellate cells, which play a pivotal role in fibrosis [42]. This suggests the intimate relationship between MMPs and fibrosis.

The enzyme activity of MMPs is regulated by a “cysteine switch” to restrict the contact between Zn^2+^ and H_2_O molecules [43]. Moreover, when the sulfide bond is broken or proteolytic cleavage occurs at the bait region, the catalytic domain is exposed and activated [44]. A well-known model of regulation is the binding of the catalytic domain by TIMPs [45]; the inhibitory process involves the 1:1 binding of MMP/TIMP on cell surfaces of cutaneous keratinocytes or fibroblasts [46]. Thus, regulation of the expression of either MMPs or TIMPs is essential for maintaining ECM balance.

### 2.3. The Connection of Expression Profile and Organs

The functions of MMPs depend markedly on their localization. Since different physiological functions of various systems may need different types of MMPs, they exhibit diverse expression ratios in different organs. Thus, we compared the RPKM (reads per kilobase per million mapped reads) of 21 MMPs (MMP-1, 2, 3, 7, 8, 9, 10, 11, 12, 13, 14, 15, 16, 17, 19, 20, 21, 24, 25, 26, 27) in 16 human tissues from 19 biosamples of Expression Atlas by Illumina bodyMap2 in NCBI GENE (https://www.ncbi.nlm.nih.gov/gene). The RPKM represents the transcript reads in the same gene length arranged by their total expression rate (see Figure 1). Notably, four of 16 organs (heart, liver, lung, and kidney) are likely to suffer from fibrosis, in which 25.64% of all MMPs are expressed (Figure 1B). Furthermore, in fibrotic organs, MMP-2, MMP-14, MMP-7, MMP-24, MMP-15, MMP-19, and MMP-9 constitute more than 90% of total MMP expression, which are reported to be key regulators of tissue fibrosis [47,48,49,50,51]. As shown in Figure 1C, MMP-2 is a pivotal MMP enzyme, except for in the brain, liver, kidney, and white blood cell (WBC). Thus, targeting these major MMPs might be harmful in other organs away from the fibrosis tissue. In mouse MMP mutant strains, there are subtle differences in the phenotypes depending on the particular mutated MMP, including reduced body size [52], obesity [53], reduced hepatic fibrosis [48], delayed mammary tumorigenesis [54], and bone development defects [55]. Based on this, members of the MMP family show good compensation for deficits of other members, except that MMP-14 alone (MT1-MMP) mutant mice show lethality and die by 3–12 weeks of age [56]. Consistent with this, our data as presented in Figure 1 showed that MMP-14 was expressed in almost all listed organs, which might explain the lethality of its mutation.

## 3. The Role of ECM Degradation in Fibrosis Treatment

Fibrosis is a disease associated with an abnormal wound-healing response in tissues such as the skin, liver, lung, kidney, and heart. To protect the site of injury and prevent infection, fibroblasts or circulating fibrocytes migrate and proliferate to secrete ECM as well as form scar tissue. The fibroblasts are activated and transdifferentiate into myofibroblasts, resulting in excessive ECM secretion. In a normal state, these myofibroblasts undergo apoptosis once the injured tissue is repaired. However, if the regulation of myofibroblast apoptosis and ECM degradation is impaired, tissue fibrosis and damage can occur. However, the key mechanisms behind this, such as the inflammatory response, elevated cytokines, and changes in the microenvironment, are still unclear.

Given that MMPs are known to function in the degradation of ECM, it was considered that they may have potential for treating fibrosis. Specific components of the ECM, such as collagens (I, III, V, and VII), fibronectin, elastin, and proteoglycans, are present at increased levels during fibrogenesis [57]. Therefore, MMPs are upregulated in the early and late stages of fibrosis in response to ECM accumulation and correlate with the fibrotic process [58].

### 3.1. Rationale for MMPs in Digesting Fibers

For digesting excessive ECM and replacing fibrosing foci with normal tissue, numerous lines of evidence have revealed that MMPs such as MMP-2 and MMP-9 increase their expression to achieve wound healing in the recovery stage [59,60], at a stage generally called re-epithelialization [61]. Some researchers have considered that fibrosis might originate from the failure of re-epithelialization, with MMPs and their functions potentially playing important roles in the Drosophila basement membrane remodeling [62]. Particularly, expression of the MMP-1 gene is known to be remarkably increased in fibroblasts in hepatic fibrosis, but not in those with liver cirrhosis [63]. It has been noticed that MMP-7 activity is measured from the serum of children with cystic fibrosis [64]. Moreover, up to a 7–12-fold increase in MMP-2 gene expression was found in in CCl_4_-induced liver fibrosis rats, whereas the 65 kDa active form of MMP-2 was enhanced 13–28-fold in comparison to that in the control group, as revealed by a zonography assay [65]. These results proved that MMPs are directly and indirectly correlated to fibrosis, but the actual roles that they play in fibrogenesis remain unknown.

### 3.2. Therapeutic Potential of MMP Inhibition but Not Activation

To determine whether MMPs play key roles, the specific knockout of genes is beneficial. As a result of such a knockout, the relationships of MMPs with fibrotic diseases in the lung, skin, and kidney have started to be revealed [66,67,68]. For instance, unilateral ureteral obstruction (UUO)-induced kidney fibrosis in MMP-9 KO mice showed significantly lesser interstitial fibrosis than in wild type mice [69]. Moreover, in the 2000s, researchers started to identify the correlation of MMP/TIMP ratio and tissue remodeling [70,71], and also evaluated the therapeutic potential of each type of MMP [72]. However, almost all MMPs have highly overlapping substrates and shared functions, as described above, so the knockout of one specific MMP gene may not successfully result in loss of function. In this context, the use of tetracycline-like antibiotics, which inhibit MMPs, were previously approved for treating infection [73]. Compared with healthy human samples, in a patient with idiopathic pulmonary fibrosis (IPF), MMP-3, MMP-9, and TIMP-1 showed decreased levels in bronchoalveolar lavage fluid (BALF), but the forced vital capacity and six-minute walking distance showed no differences [74]. Although more studies have been performed in the liver and lung than in the kidney or other organs, it was concluded that the results are often diverse for each MMP type and also differ from the animal model in the systemic analysis by Giannandrea and Parks in 2014 [18]. For example, MMP-12 has both anti- and pro-fibrotic effects in pulmonary fibrosis [75,76], and also does not affect the genetic loss of MMP-9 in rodent models [77]. Rather than diminishing fibrotic scarring, MMPs were also surprisingly shown to enhance fibrogenesis more often than improve it [78]. Surprisingly, MMPs display pro-fibrotic activity in most cases, so the use of MMP inhibitors might have potential for treating fibrosis.

In the literature, findings from studies on activation or knockout animals have indicated the importance of MMPs in the development of fibrosis [78]. However, the anti-fibrotic effects of MMPs in the lungs vary from those in organs including the liver, heart, and kidney [18]. In Table 1, we present the MMP inhibitors that have been shown to be effective in fibrosis animal models, including those with lung, liver, and myocardial fibrosis. In general, broad-spectrum MMP inhibitors are effective in all three organs. According to fewer reports on selective inhibitors, their development by pharmaceutical companies might only have been reported in patents. Moreover, some studies are not shown in Table 1 since they only involved using cell models for mechanistic analyses or for treatments having potential only at the acute injury phase [79,80]. Nevertheless, the drugs shown to have potential by animal studies still need to be tested in preclinical studies before being applied to patients.

## 4. More Functions of MMPs

Although substantial studies have revealed the functions and physiological roles of almost all MMPs, it appears that many proteins that interact with MMPs have yet to be discovered. A major clinical use of MMPs is to prevent cancer metastasis. However, studies have also utilized MMPs as targets in different body systems, including the immune, cardiovascular, and central nervous systems. Therefore, the unexpected results for fibrosis treatment may be because these additional targets also interact in fibrogenesis, such as in immunity and cell growth, and transduce messages [91,92].

### 4.1. Promotion of Cancer Invasiveness

The process of tumor cell spread from the primary site into other normal tissue, referred to as metastasis, occurs in invasive cancer via movement through the bloodstream. The surrounding tissue border and the vessel walls have to split, and typically MMPs secreted on the cell surface break the basal membrane, allowing escape into the blood flow [93,94]. Further studies showed that when the space and nutrient support is no longer sufficient for tumor growth, MMPs indirectly promote invasiveness [95].

### 4.2. Macrophage Degradation of the Basal Membrane

Macrophages play important roles in the immune system via their ability to perform phagocytosis and amoeboid movement toward wound tissue that has become infected. The activation of macrophages leads them to secrete MMPs, which is required for degradation of the ECM of the basement membrane, followed by migration into injured tissues [96,97] and the engulfment of pathogens. Macrophages from MMP-12 deficient mice exhibit abrogated migration ability [98]. In experimental studies, it was shown that the MMP-2 and MMP-9 inhibitors ARP100 and AG-L-66085 significantly reduced migration via different mechanisms: ARP100 inhibited MMP-2 and subsequent transforming growth factor β1 (TGF-β1) secretion, whereas AG-L-66085 diminished the angiogenesis response by reducing vascular epithelial growth factor (VEGF) in a retinoblastoma model [99]. A clinical trial for recurrent glioblastoma is going to test the combination of monoclonal antibody of MMP-9 combined with bevacizumab (NCT03631836); however, the concept of blocking MMP-9 is related to tumor vascularization but not to its ECM-degrading role [100].

### 4.3. MMPs Treat Stroke or Cardiovascular Diseases

MMPs also have the potential to treat diseases such as stroke, cardiovascular diseases, and arthritis [25,101,102,103]. In leukocytes, MMP-2 expression is positively correlated with the formation of sclerotic plaques, but the mechanism behind this are still unknown [104]. The above diseases are not related to degradation of the ECM, but functional components could be involved, such as NF-κB and hypoxia-inducible factor 1α (HIF-1α) [105,106]. An increasing number of additional catalytic targets of MMPs in various tissue types have also been found recently [107].

### 4.4. Central Nervous System (CNS) and the Microenvironment

Obstruction of the breakdown of extracellular constituents can cause CNS disease such as multiple sclerosis (MS); several types of MMPs are reported to be involved in MS [108]. In a Theiler’s murine encephalomyelitis (TME)-induced MS disease model, MMP-12 was shown to play a pivotal role in the development of astrogliosis and demyelination [109]. These processes are similar to fibrosis, including a collagenous region and focal proliferation in brain tissue. MMP-12 was also shown to reduce activated microglia and reactive macrophages, influencing the M1/M2 balance in virus-infected mice [110]. The therapeutic mechanisms include the macrophage-mediated proteolysis and matrix invasion and the basement membrane penetration potential of macrophages [111].

Recently, it is reported that MMPs regulate the microenvironment via the shedding of the exosome [112]. The exosome is a small releasing vesicle (30–100 nm in diameter), the trafficking of which enables communication with other cells and the movement of cargo such as proteins, cytokines, and miRNAs [113]. It is closely related to the physiological and regulatory mechanisms of the exosome; thus, MMP shedding for appropriate release plays a role in mediating their functions. As such, adipocyte-derived exosomes of the liver were shown to induce TGF-β signaling in hepatocytes, leading to the initiation of fibrosis [114]. Moreover, exosomes containing MT1-MMP activated pro-MMP-2 and caused the subsequent degradation of type 1 collagen and gelatin in the fibroblast-like cell line COS-1 [115]. Furthermore, the enzymatic activity of MMP-2 is involved in exosome trafficking from fibroblasts to endothelial cells and facilitates the breakdown of ECM in an MMP-14-dependent manner [116].

## 5. Possible Participating Role of MMPs in Fibrosis

In terms of the possible mechanisms of MMP involvement in fibrosis, the first candidate is the inflammatory response after injury [117,118], which was utilized as one of the clinical drugs for treating lung and liver fibrosis [119,120]. Anti-inflammatory therapy has been proven to have potential in animal models; however, in a clinical trial, treatment with steroids such as prednisolone in patients with pulmonary fibrosis resulted in unfavorable outcomes and increased hospitalization events [121]. Some researchers have been shifting their focus away from immunology and fibrosis, but the pathological relationship remains [122]. Notably, the correlation between fibrosis and MMPs in macrophages and neutrophils [123], such as MMP-1, MMP3, MMP-7, MMP-9, MMP-13, and MMP-19, was shown to be increased in BALF. These infiltrated macrophages and neutrophils refer to the inflammatory response and are essential in pulmonary fibrosis [50,71,124,125,126,127].

### 5.1. Immunomodulation or Inflammatory Regulation

Additionally, MMP-10 is known to be related to the transition from the M1 to M2 type of alveolar macrophages and regulates the immune tolerance to TLR-7 induced inflammation [128]. Besides the inflammatory effect, the following wound healing response usually includes peripheral epithelium activation through the EMT in renal tubular epithelial, hepatic stellate, and alveolar epithelial cells [129,130,131,132]; these cells transdifferentiate into fibroblast-like or myofibroblast cells and secrete collagen fibers, leading to fibrosis [133,134]. Accordingly, MMPs contribute to EMT-related cancer metastasis in breast and gastric cancers [21,135], although we mentioned their anti-cancer potential in the previous section. In the final stage of wound healing, the recruited myofibroblasts undergo apoptosis and allow tissue regeneration, whereas in fibrosis, the myofibroblasts are resistant to apoptosis and proliferate to form fibrotic foci [136].

The pro-apoptotic role of MMP-7 is due to the cleavage of CD95 in apoptosis-resistant tumor cells [137]. Moreover, MMP-10 promotes tumor progression by stimulating HIF-1α and MMP-2 in cervical tumors [138]. Notably, MMP is needed and would be increased in tissues in cases of fibrosis; however, the ECM does not decrease in these circumstances. Therefore, MMPs are strongly suspected to interact with pro-fibrotic factors, such as molecules involved in inflammation, EMT, and apoptosis resistance (e.g., TGF-β1, IL-1β, and TNF-α) [139,140].

### 5.2. ECM and Vasculature in Angiogenesis

Vessel walls have an ECM-containing, three-layered structure, including the tunica intima, tunica media, and tunica adventitia, and provide mechanical strength and elasticity. Neoplastic proliferation requires angiogenesis in malignant or benign tumors; the existence of MMPs and VEGF facilitates tumor growth [141]. Moreover, neutrophilic MMP-9 acts as a pro-angiogenic proteinase, as revealed using a developing chicken embryo model [142]. Studies in the adventitia layer revealed that MMP-2 activation and fibroblast proliferation both induce a phenotypic switch in a hypoxic state, converting fibroblasts into myofibroblasts, which often form a fibrotic focus [143].

## 6. Conclusions

Since the complete mechanisms of fibrosis remain a mystery, the origin and each stage are a complex and mutual effect. To decrease the excessive ECM in fibrotic tissue, the activated MMPs interact with the molecules involved inflammation, EMT, and apoptosis [144]. Unfortunately, the complete functions and substrates of MMPs have not yet been revealed [145,146], although some of them are secondary messengers, such as the cleavage of the AMP-activated protein kinase-α (AMPK-α) by MMP-9 in Toll-like receptor 4 (TLR4) signaling [105]. Although studies have suggested that simple inhibition of the expression of an MMP is insufficient to treat fibrosis, we aim to clarify the roles of MMPs in fibrosis in more detail to increase the potential for using them in a clinical context in the future.

## Figures and Tables

**Figure 1 molecules-24-04188-f001:**
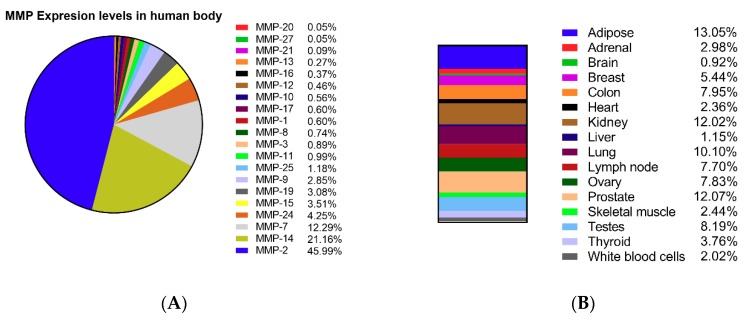
Expression of 20 types of matrix metalloproteinases (MMPs) (MMP-1, MMP-2, MMP-3, MMP-7, MMP-8, MMP-9, MMP-10, MMP-11, MMP-12, MMP-13, MMP-14, MMP-15, MMP-16, MMP-17, MMP-19, MMP-20, MMP-21, MMP-24, MMP-25, and MMP-27) in 16 human organs (adipose, adrenal, brain, breast, colon, heart, kidney, liver, lung, lymph node, ovary, prostate, skeletal muscle, testes, thyroid, and white blood cells). The reads per kilobase per million mapped reads (RPKM) value compares the gene expression with the sample sequencing depth and gene length. (**A**) The expression ratio of different MMP types in human organs. (**B**) The distribution ratio of (**B**) all MMPs and (**C**) seven major MMPs in 16 organs.

**Table 1 molecules-24-04188-t001:** Experimental evidence of effective MMP inhibitors against fibrosis in animal models including of lung, liver, and myocardial fibrosis.

Compound Name	Description	Effects	CAS Number	Refs
Batimastat	A broad-spectrum MMP inhibitor	Inhibit pulmonary fibrosis	130370-60-4	[81]
CL 82198 hydrochloride	A selective inhibitor of MMP-13	Blocks liver fibrosis	307002-71-7	[82]
CP 471474	An MMP inhibitor	Inhibit collagen in myocardial fibrosis	210755-45-6	[83]
Doxycycline Hyclate	An antimicrobial tetracycline that acts as an inhibitor of MMP-1, MMP-8 and MMP-9	Attenuated pulmonary/myocardial fibrosis	24390-14-5	[84,85] [74]
		Reduced parameters in IPF patients		
GM 6001	A cell permeable MMP and fibroblast collagenase inhibitor	Reduced pulmonary inflammation and fibrosis	142880-36-2	[86]
Marimastat	A broad-spectrum MMP inhibitor and selective TACE inhibitor	Aggravates liver fibrosis	154039-60-8	[87,88]
PD166793	A potent MMP-2, MMP-3, and MMP-13 inhibitor	Retardation of age-associated arterial fibrosis	199850-67-4	[89] [90]
		Reduced myocardial fibrosis		
Thiorphan (DL)	An enkephalinase and metalloproteinase inhibitor	Reduced myocardial fibrosis	76721-89-6	[80]

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
