# Peer review of "The Versatile Role of Matrix Metalloproteinase for the Diverse Results of Fibrosis Treatment"

_molecules, 2019, doi:10.3390/molecules24224188_

Round 1

Reviewer 1 Report

Chuang et al. present a review of the role played by MMPs in fibrotic processes in various organs/cells/tissues and human and animal models.  The work is comprehensive and detailed, but the presentation is difficult to follow for a variety of reasons.  I have the following suggestions.

Figure 1 appears sideways and is very difficult to read in general.  The authors need to greatly improve this figure so the readership can interpret it.

The authors need to place all abbreviations in a table to assist the readership, as there are far too many to keep track of based on definitions in text and the brief paragraph provided at the end of the article.

While a time-consuming effort, an illustration or two that includes organs and specific sites of action of particular MMPs would be of great assistance to the readership.

A brief table summarizing what has been determined with knockout animal models (poor healing, lethality, etc.) would be helpful.

Author Response

Chuang et al. present a review of the role played by MMPs in fibrotic processes in various organs/cells/tissues and human and animal models.  The work is comprehensive and detailed, but the presentation is difficult to follow for a variety of reasons.  I have the following suggestions.

 Figure 1 appears sideways and is very difficult to read in general.  The authors need to greatly improve this figure so the readership can interpret it.

We have thought about how to present more clearly and thus we split the figure into 3 parts of it. The first part (A) is to identify the expression levels of each MMP types over the human body. The second one (B) goes to represent the relative MMP contents in 16 organs. The third part (C) compared relative contents of 7 types of MMP in 16 organs.

 The authors need to place all abbreviations in a table to assist the readership, as there are far too many to keep track of based on definitions in text and the brief paragraph provided at the end of the article.

Thanks for your advice, we placed an abbreviation table before the introduction.

 While a time-consuming effort, an illustration or two that includes organs and specific sites of action of particular MMPs would be of great assistance to the readership.

 A brief table summarizing what has been determined with knockout animal models (poor healing, lethality, etc.) would be helpful.

It is indeed very important to summarize the information; thus, Giannandrea and Parks have a detailed list and description with animal models. We have mentioned in Line 51~52.

Reviewer 2 Report

In this article, Chuang, Chen and Harn, review the role played by matrix metalloproteinases in the diverse results obtained after fibrosis treatment.

This is an interesting current topic, and a review that provides the reader with a comprehension overview of the field of worthy of publication. The English of this article is good, and there is only very few mistakes.

However, this article currently suffers from a sever problem of clarity !! As a reader, it is very hard to follow and understand the authors. As a result, this review, as is, would not help scientific community understanding better this topic. A showcase example of this lack of clarity is the section 2.1 (see in the specific comments below).

In addition, it is very hard to for the reader to get a global picture (while this should be the aim of a review). A huge effort should be put in rewriting the introduction and first chapters, describing first the global picture, the generalities, and then the specificities.

In conclusion, as a reviewer I recommend major revisions before accepting this article for publication. The author must modify the article extensively in order to gain in clarity (mostly from the introduction to the chapter 4 included). Plus, given the title, I would encourage the author to put a particular emphasis on the description of this “dual role” of MMP on fibrosis.

After these general comments, here are more detailed comments :

First thing to change : the figure ! It is not readable ! What is the use of a figure if the reader can’t read any label ?! And the beginning of the title is missing !

l15 : Specify “potential target” for what (for example l45-47 is very clear on that)

l16-17 : In these “various biological process”, the activity of MMP is still degrading ECM, no ? (like tumor propagation, etc) Therefore it is not correct to write “MMP participates not only to ECM degradation but…”

l47-49 : please clarify/explain more this sentence.

l50 : The end of this sentence refers to the “invasion” of what ? Cancerous cell ?

Section 2.1 is not clear at all ! (eg l75-77). In addition, the chemical mechanism as described currently is wrong ! My suggestion is : either correct and re-write it, or tell the reader in which reference an accurate description can be found.

l138 : “are produced at higher rates” or “are present at increased level” ?

l143 : proteins do not “increase their expression” but rather : “their expression is increased”

L155 : it could be mentioned here on which model/species this work has been done.

l170 : “MMPs were” instead of “was”

It is not obvious why are sections 3.2 and 3.3 separated. Why couldn’t they be merged and discussed together ? (eg their beginning is very similar)

l221-222 : this comment should be added also when describing the general roles and activities of MMPs.

l229-233 : please clarify/explain more.

L277 : the term “molecular mechanisms” usually do not designate the actual, physical molecules, but the conceptual mechanism. Therefore, MMPs cannot “interact with”. But they could interact with “the molecular components”, or the “molecules involved in inflammations”.

Author Response

In this article, Chuang, Chen and Harn, review the role played by matrix metalloproteinases in the diverse results obtained after fibrosis treatment.

This is an interesting current topic, and a review that provides the reader with a comprehension overview of the field of worthy of publication. The English of this article is good, and there is only very few mistakes.

However, this article currently suffers from a sever problem of clarity !! As a reader, it is very hard to follow and understand the authors. As a result, this review, as is, would not help scientific community understanding better this topic. A showcase example of this lack of clarity is the section 2.1 (see in the specific comments below).

In addition, it is very hard to for the reader to get a global picture (while this should be the aim of a review). A huge effort should be put in rewriting the introduction and first chapters, describing first the global picture, the generalities, and then the specificities.

 In conclusion, as a reviewer I recommend major revisions before accepting this article for publication. The author must modify the article extensively in order to gain in clarity (mostly from the introduction to the chapter 4 included). Plus, given the title, I would encourage the author to put a particular emphasis on the description of this “dual role” of MMP on fibrosis.

Thanks for your attentive review and generous comment. We have rewritten most sections especially the introduction part to emphasize the importance and possible reasons. The revises to each comment are listed below:

 After these general comments, here are more detailed comments :

 First thing to change : the figure ! It is not readable ! What is the use of a figure if the reader can’t read any label ?! And the beginning of the title is missing !

We have thought about how to present more clearly and thus we split the figure into 3 parts of it. The first part (A) is to identify the expression levels of each MMP types over the human body. The second one (B) goes to represent the relative MMP contents in 16 organs. The third part (C) compared the relative contents of 7 types of MMP in 16 organs.

 l15 : Specify “potential target” for what (for example l45-47 is very clear on that)

We revised it into “Thus, matrix metalloproteinases (MMPs) are one of the main groups of ECM-degrading enzymes, and they are the potential target of fibrosis treatment.”

l16-17 : In these “various biological process”, the activity of MMP is still degrading ECM, no ? (like tumor propagation, etc) Therefore it is not correct to write “MMP participates not only to ECM degradation but…”

We revised it into “However, MMP participates not only in ECM degradation but also in development and various biological processes such as the potential to treat diseases such as stroke, cardiovascular diseases, and arthritis.” And this also replied to the comment for L221-L222.

l47-49 : please clarify/explain more this sentence.

We revised it into “Although most studies have shown that disruption of the activity or expression of MMPs reduced fibrosis, Giannandrea1 and Parks listed the diverse treatment results for fibrosis in different types of MMP”

l50 : The end of this sentence refers to the “invasion” of what ? Cancerous cell ?

We revised it into “Moreover, the contradictory roles of MMPs have been reported not only in the tumor microenvironment but also in relation to the acquisition of properties for cancer growth and invasion.”

 Section 2.1 is not clear at all ! (eg l75-77). In addition, the chemical mechanism as described currently is wrong ! My suggestion is : either correct and re-write it, or tell the reader in which reference an accurate description can be found.

Thanks to your generous advice, we both re-wrote the section and provided the reference for detail study.

 l138 : “are produced at higher rates” or “are present at increased level” ?

We revised it into “Specific components of the ECM, such as collagens (I, III, V, and VII), fibronectin, elastin, and proteoglycans, are present at increased level during fibrogenesis.”

l143 : proteins do not “increase their expression” but rather : “their expression is increased”

We revised it into “numerous lines of evidence have revealed that MMPs such as MMP-2 and -9 are increased their expression to achieve this in the late stage of wound healing.”

L155 : it could be mentioned here on which model/species this work has been done.

We added the following description “For instance, unilateral ureteral obstruction (UUO)-induced kidney fibrosis in MMP-9 KO mice showed significantly less interstitial fibrosis relative to the wild type mice.”

l170 : “MMPs were” instead of “was”

We revised it into “MMPs were also surprisingly shown to enhance fibrogenesis more often than to improve it.”

 It is not obvious why are sections 3.2 and 3.3 separated. Why couldn’t they be merged and discussed together ? (eg their beginning is very similar)

We merged sections 3.2 and 3.3 and omitted some unnecessary parts, the sentences are smoother than before. It really helps.

 l221-222 : this comment should be added also when describing the general roles and activities of MMPs.

We added in Line 17~18 of the abstract part.

l229-233 : please clarify/explain more.

We added “Recently, it is reported that MMPs regulate microenvironment via the shedding of the exosome.”

L277 : the term “molecular mechanisms” usually do not designate the actual, physical molecules, but the conceptual mechanism. Therefore, MMPs cannot “interact with”. But they could interact with “the molecular components”, or the “molecules involved in inflammations”.

We revised it into “To decrease the excessive ECM in fibrotic tissue, the activated MMPs interact with the molecules involved inflammation, EMT, and apoptosis.”

Reviewer 3 Report

This is an updated and well-written comprensive aper on this important

and complicatedinterdiscplinary

and translational medical /biotechnological topic.

Recommended for publication.

Author Response

Thank you for your recommendation. We have a revised version and hoping that would be better.

Round 2

Reviewer 1 Report

Thank you for the new illustrations.  They greatly enhance the work.

Author Response

Thank you for your advice.

Reviewer 2 Report

The authors took into consideration the comments or reviewers. In particular, the correction of Figure 1 is a significant improvement of the paper, the authors did a good job on that.

By modifying the text, the authors introduced mistakes (mostly english). Therefore, please correct the following lines:

L15 L16-18 L162-163

However, the description of the catalytic mechanism, as described, is still chemically incorrect. This is a major issue. Therefore, I suggest either the authors ask the help of a chemist to correct it, or, they completely remove this part from the review and send the readers to other papers/reviews.

After that, I would recommend the paper to be published in Molecules.

Author Response

Thank you for your comments, we have deleted our descriptions on the catalytic mechanism. Moreover, grammar mistakes are revised again through the article by our English editor in Lines 15, 16-17, 51, 162-163, 175-176, 285, and 288-289. 

Hope that would be better.